# BraveNet Upstander Social Network against Second Order of Sexual Harassment

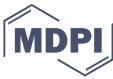

**Lidia Puigvert** [1,*] , **Ana Vidu** [2] , **Patricia Melgar** [3] **and Marifa Salceda** [4]

1 Department of Sociology, Faculty of Economics and Business, University of Barcelona, 08034 Barcelona, Spain
2 Department of Private Law, School of Law, University of Deusto, 08007 Bilbao, Spain; ana.vidu@deusto.es
3 Department of Education, School of Education, University of Girona, 17004 Girona, Spain; patricia.melgar@udg.edu
4 Department of Education, School of Languages and Education, University of Nebrija, 28015 Madrid, Spain; msalceda@nebrija.es
* Correspondence: lidia.puigvert@ub.edu

**Abstract:** Gender-based violence and domestic violence constitute a huge problem all across countries and continents. The COVID-19 outbreak and the lockdown produced as a consequence of it have contributed to escalating this problem. Many national organisms reported an increase in the data on domestic violence during confinement. Bystander intervention often constitutes one of the most effective mechanisms of attention. The problem is that bystanders do not always dare to intervene. This article aims to provide knowledge on the reasons for this lack of intervention and its connection to domestic violence, while presenting measures to encourage intervention and victim support, offering protection to those most in need during this pandemic. The research was conducted through questionnaires distributed online among social entities in charge of providing care to women suffering from domestic violence during the lockdown. The results have shown that most of these entities have had to intervene in providing support to women during the lockdown. In conclusion, the case of the Unitary Platform Against Gender Violence and the entities, which are members of the platform, acted in situations of domestic violence produced during confinement, based on the mutual support provided by being a group of entities that have the support of the Platform.

**Keywords:** domestic violence; gender-based violence; lockdown; support network; COVID-19

## 1. Introduction

Gender-based violence is a problem affecting women and children all across countries and continents. Data from ONU [1] has shown one in three women around the world were victims of some type of gender-based violence (GBV). In December 2019, the COVID-19 outbreak caused health, economic, and social crisis unprecedented in our century. People infected with COVID-19 were growing and growing all around the world [2], and lockdowns started as a response in different countries as the pandemic uncontrollably spread and usual activities dramatically changed. The WHO reported so far, as of 6 April 2021, 131,487,572 confirmed cases of COVID-19, including 2,857,702 deaths, all around the globe [3].

At the beginning of March 2020, a wave of lockdowns started to quickly be the reality of several countries and current people's activities suddenly changed. Even though data is still limited, considering the United Nations report of May 2020 [4], Member States have reported up to a 60% increase in emergency calls from women subjected to violence by their partners in April 2020, compared to the same date of the previous year. Other organisms such as the World Health Organization promptly published useful information on violence against women during COVID-19 [5], including measures and advice on how to be safe at home, as the awareness existed that homes are not a safe place for everybody [6].

In March 2020, coinciding with the beginning of strict confinement in many countries, the GREVIO (Group of Experts on Action against Violence against Women and Domestic

Violence) representative, mechanism of the Council of Europe in charge of implementing the Istanbul Convention [7], asked for building solutions and alternatives for women and children who may hardly suffer from domestic violence during lockdown occasioned by COVID-19 [8]. In fact, domestic violence has escalated since the COVID-19 outbreak. For instance, in the UK, murder cases against women have been doubled since the pandemic broke out [9]. In April 2020, the Executive Director of UN Women informed similar data on violence against women and girls [10]. To mention some cases, in Cyprus, France, and Australia, reports of domestic violence have increased 30%, in Argentina the number increased to 25% and in Singapore to 35%. In countries such as Canada, Germany, Spain, and United States, among many others, demands for emergency shelter have arisen [11].

Home might have been a solution to stop spreading the virus, but there are no doubts that, for many women and children, home is just not a safe place, as the Council of Europe informed [8]. The impact of the COVID-19 crisis has significant transcendence on women, who stand at the core of the fight against coronavirus [12]. Indeed, the consequences of COVID-19 on domestic violence have been even defined as the pandemic paradox, considering a scenario in which the global pandemic produced unintended negative consequences, especially for those already vulnerable as women fall victim to domestic violence [13]. While it is true that domestic violence may affect all genders, according to the data, more often it affects women, and this was mainly the case during the lockdown.

Facing this reality, several people, including famous and well-known persons, decided to make important donations in order to help women and children victims of domestic violence (DV). This is the case of Charlize Theron [14] who calls for "other influential women and organizations to join this critical cause in providing safe spaces and rescue programs for women in need". However, the broad question still states why those who have the position of supporting survivors, do not always do so. Everybody is aware of women's rights and especially women's rights and the COVID-19 pandemic, as the Council of Europe states. What we attempt to bring to the DV spectrum goes in line with the document released by the European Commission on the need to protect those who claim "Better protection of whistle-blowers: new EU-wide rules to kick in in 2021" [15].

### 1.1. Networks of Support and Bystander Intervention as Ways of Addressing Domestic Violence and SOSH during Lockdown

In order to eradicate domestic violence, it is necessary to generate wide social support that helps victims to report, feel accompanied, and protected throughout the process. When intervening, victim's supporters are often discredited and suffer attacks, which re-victimize the victim by weakening their support network and increasing the risk of isolating them. This reality is defined as the Second Order of Sexual Harassment (SOSH) [16]. Wondering to what extent this situation is reproduced during confinement, whether bystanders intervene, and under which conditions, and who are the ones supporting, constitutes the aims of this paper.

This article undertakes the second order of sexual harassment, as such harassment received by people who support survivors take the risk to suffer reprisals because of that [17]. The recent inclusion of SOSH in the legislation on violence against women by unanimity in the Catalan Parliament [18], means both the public recognition of this reality as well as progress in the eradication of violence against women. This is the first known legal framework which has incorporated the second order of sexual harassment.

As research has shown, bystander intervention [19] is considered one of the most effective intervention mechanisms. Researches on actions and mechanisms for reporting sexual harassment show that, besides its good results, it has not implied a significant reduction of the incidence of sexual harassment over decades of implementation. The issue of bystander intervention emerged and they do not always dare to intervene, as they might feel fear of retaliation and reprisals against them and their loved ones [20]. This limits the victim's support, which is crucial for deciding to keep forward or to complain.

Social support is crucial for victims to overcome and reduce the impact of the situations they suffer [21]. Families of domestic violence need intervention, bystanders who dare to

intervene [22]. Children also may suffer as indirect victims because of witnessing domestic violence often suffered by their mothers [23]. The ones embodying disbelief are often adults surrounding pupils, within the family, neighborhood, or even in school. To end this whole problem, which is also happening within the familiar environment, support networks for them and external implication are definitively needed [24]. The whole community has to protect and be protected [25].

The concept of Second Order of Sexual Harassment (SOSH) was first used in the academic context [26] to describe a reality that became aware years after concrete anti-harassment measures had been implemented in different universities. The fact that pushed it consists of realizing that victims still did not dare to report incidences despite counting on formal mechanisms. Dziech and Weiner [26] state: "Sexism on campus creates a second order of sexual harassment victims, those who advise, support, and rule in favor of the primary victims. These are the affirmative action officers, ombudspersons, counselors, assistant deans -the people assigned, and usually committed, to helping sexual harassment victims". Further research has focused on the effects on direct victims, and thus, on the need of empowering them [27], based on concepts such as "survivors first" [28]. However, especially because survivors need to be supported, their supporters have also to receive protection.

### 1.2. Approaching SOSH as a Way to Face and Prevent Attacks

Moving from direct victims to second order victims, as mentioned, the concept of second order of sexual harassment victim contemplates any person who becomes a victim (and suffers from what this category entails) when they decide to support, help, or position themselves with a direct victim. Support for second order victims should be enforced by any legislation that recognizes the need for victims to be supported as well as for SOSH to be considered. To that extend, supporters need to feel safe. The manuscript considers the protection for those active bystanders and the potential harassment they may receive because of that.

The consequences suffered by second order victims of gender-based violence at some point may be similar to the ones suffered by direct victims, including physical and/or psychological effects [29]. The reality of the second order of sexual harassment focuses on considering those people who support direct victims as potential victims (of second order) because of their active support. This study aims to demonstrate the relevance of bringing SOSH into the domestic violence spectrum that occurred during COVID-19 confinement.

Tackling the second order of sexual harassment raises awareness not only on protecting direct victims but also those who dare to support them. Therefore, it is especially relevant to approach SOSH in the pandemic context to provide a scientific explanation for the increasing domestic violence and to argue on the need of intervention. Social movements, survivor activists, are making their voices heard leading the struggle that men and women carry on for decades, standing always on the victim's side, not looking the other way; even suffering hard consequences for doing so [29].

## 2. Methods

For the present research, the authors conducted a statistical analysis of data provided from responses of entities' representatives to a questionnaire specifically designed for this purpose. Data were analyzed following the quantitative methodology.

### 2.1. Participants

The participants in this research were social entities (n = 98), different kinds of entities with different aims, but with the common focus that all of them are official members of the Unitary Platform against Gender Violence [30]. We choose to study the case of this Unitary Platform as being a unique sum of entities, huge in its region, and serving its scope and functioning for decades. The Unitary Platform has been founded in 2002 to provide response to the need to make gender-based violence visible and to demand action through

citizen mobilization. The Platform is made up of individual entities and a large group of volunteers, professionals, and other people sharing this Platform's values. Entities are from all over the Region of Catalonia, in Spain, and they work on promoting a social movement with the mission of eradicating violence against women. This article is a case study of this region and the particular entities included in the survey.

In order to find the entities associated with the Unitary Platform, we had a look at the Platforms' webpage, in the section "entities and assembly". We found a list there of 130 entities associated with the Platform (121 of which have membership status). Then we elaborated our own excel and started to search the webpage of each and every one of these entities. Finally, we found an available connection with 98 entities. A member of our research team sent the questionnaire (in a google drive format) to the official email address of each of these entities, separately. A representative of each entity answered the questionnaire providing information about the entity case and mechanisms against domestic violence. They were the ones deciding whom to respond to in the questionnaire. As researchers, we actually do not know the identity of the ones responding in order to maintain anonymity. Our aim was to gather information regarding their actions during the lockdown produced by the COVID-19 outbreak to help women who were suffering domestic violence.

Finally, we decided to use a total of 23 questionnaires (n = 23) for our study. There were no missing values among the variables that were chosen for the analysis. To mention some profiles, most of the entities are associations specialized in caring for victims of domestic violence, some specializing in protection and prevention, some are associations of adults, and some are research groups dedicated to research in overcoming gender violence. There are also student associations, women's legal associations, journalists associations, migration and economic affairs associations, trade unions, among other entities. Based on their diversity, all entities of the Unitary Platform are dedicated to improving the situation of women victims of gender-based violence.

*2.2. Instrument and Measures*

The instrument used for this research was a questionnaire designed by the researchers of this study, based on previous knowledge on GBV [31] and quantitative methodology of research, using batteries of questions aimed at measuring the incidence of second order of sexual harassment regarding the gender-based violence produced during confinement in the context of COVID-19. As research shows [29] the way of dealing with or preventing attacks makes a difference in the support and approach of gender-based violence. As objectives of the questionnaire, we ask the following questions: What has happened during the confinement, which people or entities have intervened in cases of DV, and which impact their interventions have suffered. In addition, an important goal of the instrument consists of collecting data on possible retaliation for intervening that may have occurred, that is, on the incidence of second order of sexual harassment that potentially could be given during confinement to the interventions of the community or associations and entities in situations of domestic violence. We expect this analysis to help ensure that the most successful actions during this period could also be transferred to other contexts, or to similar interventions in a post-pandemic period.

The questionnaire contains four blocks that can be answered in approximately 10 min. Block 1 includes sociodemographic questions of the corresponding entity; exploring their specialization, how long they have been members of the Platform, what their participation in the Platform consists of, and their potential participation in other entities. In Block 2, the questions focus on gathering knowledge about having defended the victims during confinement, inquiring if they have received any kind of attack, if they knew other people who had also intervened, as well as what kind of support or discouragement they received by other people or groups. Block 3 focuses on collecting information on the ways they use to support victims, actions, and specific mechanisms implemented. Block 4 focuses on the case and pathways of the Unitary Platform against gender violence; and how entities

perceive the fact of being part of this Platform in relation to their intervention and their support mechanisms.

### 2.3. Analysis

The analysis was made following two realms, on one hand, we focused on the data collected from the questionnaire; and on the other hand, we focused on the existing theory on domestic violence, survivor support, and consequences of bystander intervention. More concretely, two phases were implemented during the analysis.

In the first phase, we applied the descriptive analysis of the indicators used arguing the reasons we followed for addressing each question of the different sections of the questionnaire. In the second phase, we tested the intervention process followed by the entity members of the Platform during the confinement and the DV cases that occurred during that period.

The indicators we used to analyze and interpret the results are the following:

1. Entity scope. We inquired about the main role of the entity in terms of DV intervention, as well as their concrete scope in addressing GBV;
2. Intervention in confinement. As the COVID-19 pandemic produced a new situation for everybody, very challenging especially for women suffering from violence, we strove to know how entities intervene and under which circumstances;
3. Potential attacks for intervening in supporting survivors. Testing the second order of sexual harassment presence in cases of intervention during the lockdown;
4. Reasons to support survivors. Which elements contribute to encouraging some entities to intervene in supporting survivors, despite the COVID-19 crisis and besides potentially suffering attacks for interviewing? Inquiring potential people who do not intervene or discourage intervention;
5. Unitary Platform membership. In this sense, we researched to what extent being a member of the unitary platform implies a powerful support network.

Differences between the confinement situation and the previous intervention in the case of domestic violence were also studied.

### 2.4. Ethical Consideration

Following the ethical rigors of the international scientific community pursued by the research team to conduct this research, the questionnaire was anonymous and the personal data of those who completed it were anonymously analyzed and will not be disclosed anywhere.

The distribution of the consent form was conducted in two ways. On the one hand, it was incorporated in the questionnaire as "block 0", so that people had to click "ok" and "consent" to their participation in order to continue in the following blocks filling out the rest of the questionnaire. On the other hand, a sheet with information about the research and informed consent for participation in the study was sent to all participants attached with the email. People also had the option of filling out the informed consent form and sending it back to the research team. Consent could be signed electronically or by hand with a scanned copy emailed back to us. In these cases, through informed consent, the people collecting the data knew who had participated, but it was guaranteed that the anonymity of their participation and their responses would be maintained for the rest of the researchers and for everyone. Indeed, they were informed that no one else would know any data but the person in charge of managing the consents and a member of the research team.

Participants were also informed that the questionnaire was voluntary, so no entity should feel compelled to complete it. The analysis does not include any type of personal data, nor any name so that not even the entity could be identified. They were also informed that the data would be used exclusively for this research. The information was used in accordance with current data protection legislation [32]. The research was sub-

mitted for evaluation by the CREA Ethical Committee and approved under the reference code 20210228.

At the same time, confidentiality and ethical issues have been carefully cared for during all the processes of creation, distribution, and analysis of the instrument. EUvsVirus-SOSH were drawn upon ethical procedures defined by the EU's Charter of Fundamental Rights and the UNESCO Universal Declaration of Human Rights.

## 3. Results

Approaching to what extent the entity members of the Unitary Platform have intervened in cases of DV, before and during the confinement, and also whether they have suffered attacks when intervening and how they have overcome them, a questionnaire was launched and analyzed, contributing to a new model in line with its results.

### 3.1. Entity Scope

In relation to our first element of analysis, we observed that 52.2% of the entities surveyed are specialized in dealing with cases of sexist violence. Among the remaining entities, some of them are dedicated to violence prevention, awareness-raising, legal advice, or other types of victim care. There is no entity in our survey whose main objective is not related to deal with domestic violence in one form or another. A total of 72.8% of the entities surveyed have been part of the Unitary Platform for more than five years, and 36.4% for more than 10 years. Among the entities, 27.3% have been members for less than five years, which shows that there are relatively new entities on the Platform, and above all, the diversity among all of them shows the richness of the Platform. In fact, 69.6% of the entities also declared to be part of some other network of entities or platforms.

### 3.2. Intervention in Confinement

During the confinement, 56.5% of the entities surveyed stated that they had intervened by giving direct support to a victim of domestic violence at least at some point. In addition, 69.6% knew other people who have done it. In fact, 60.9% of the people surveyed claim to have developed some type of specific action or campaign to promote support for victims in that period. These types of activities are related to direct care, support, or/and accompaniment. Actions are also training awareness, dissemination of services, attention, and accompaniment by phone or via WhatsApp. There have also been manifestos elaborated by some entities or political requirements. These might be considered conditions under which bystander intervention is managed during confinement.

In relation to the fact of encouraging others to intervene, 78.3% affirmed that, during confinement, no one told them that it was better not to get involved. Instead, according to 21.7%, on other occasions, they had been encouraged not to intervene. This fact could inspire to understand that the confinement situation is an exceptional occasion in which the number of people who discourage others from intervening is low. As a gratifying result of the intervention, 38.1% of the entities state that during the confinement the victims or their relatives thanked them for their involvement. While 52.4% affirm that during the confinement they did not receive acknowledgments, but they did on previous occasions. This concludes that during confinement different aids are required, and a slightly different reaction on the part of the people also helped.

### 3.3. Potential Attacks for Intervening in Supporting Survivors

When we asked the entities about any type of attack, offense, or criticism that they may have received for supporting a victim during confinement, 68.2% answered negatively. A total of 31.8% answered that during confinement they did not receive attacks, but they did receive them on other occasions. Of the people who acknowledge having suffered attacks for intervening, 50% of those surveyed said they had no support to face these attacks. No one answered affirmatively to the question, so no one received attacks for intervening during confinement. Given that, some of the responses state that they have

received attacks prior to confinement, but not during that period. We could conclude that the incidence of SOSH decreases in confinement. Another possible interpretation of these data leads us to take into account that little identification of the second order of sexual harassment still exists [16], which may imply that some entities entirely do not identify a recently legislated reality, although in many cases it exists in their own contexts. Regarding bystander intervention, even if it might be of different kinds (by friends, neighbors, a family of the victim, or rather formal members of some institutions), the paper focuses on the attacks and reprisals those people might receive, to raise awareness and prevent it in order to overcome the problem of violence.

Besides potential attacks for interviewing, the entities shared different reasons that contributed to encourage some of them to intervene in supporting survivors, despite the COVID-19 crisis, and besides potentially suffering attacks for interviewing. These elements are based on mutual support, having a clear position, always on the side of the victims; having a clear response and feminist action to an attack or from the solidarity organization among the very best that is part of it.

### 3.4. Unitary Platform Membership

From all the surveys conducted, when asked about their personal intervention in attending a potential survivor of domestic violence, almost half of the people surveyed affirmed that during confinement, the fact of being part of the Unitary Platform, and having the chance to feel supported by a network as the Platform, empowered them to intervene and take a position on the side of the victim. Indeed, networks of support encourage intervention and make people feel accompanied. From the fieldwork, we also realized that through their responses, the entities also identified that being members of this Unitary Platform was crucial, especially during confinement. This membership contributes to increase the support they may have had in case of need. Thus, in the event of possible attacks for taking the victims' side, they would count on supportive people.

The entities state that some of the examples in which this support has been noted are expressed through expressions such as the following: "to get to know each other and rely on everything", "to know that they can count on each other", "to have the actions of others as references". Besides the closed questionnaire questions, we also provide some space for people to freely write down whatever they considered in regards to how important it is for them to be a member of the Unitary Platform. For instance, one participant wrote on an open question of the questionnaire: "We know that we are part of a group that does not tolerate any type of violence and that is positioned in cases of second order of sexual harassment, this provides us the confidence of knowing that when we act, we will have the chance to count on them."

### 3.5. BraveNet 0 Violence: Upstander Social Network

Since the support networks are the key to intervene, since the Platform entities, in their great majority, intervene because they have the support of the Platform and/or are also members of some other support platform, a support network is crucial, so our proposal is the BraveNet upstantder social network. This proposal responds to the problem of the increasing DV since COVID-19 outbreak in different countries and continents. The BraveNet network was elaborated during the EUvsVirus Hackathon [33], celebrated in April 2020, to create solutions to the different problems caused by the COVID-19 crisis. It is clear that DV needs upstanders in order to be eradicated. Bystanders do not always dare to intervene, mainly because of suffering reprisals, attacks, and negative consequences. The fact of intervening makes them move from bystanders to upstanders; or from someone who witnesses a violent act, or is aware of it, to someone who acts against it or avoids it from happening. Some upstanders become victims of the second order of sexual harassment because of supporting direct victims. Indeed, their role is crucial to uncover DV in confined situations. If there were no upstanders, victims would have enormous trouble breaking the silence as evidence shows [29,31]. The pioneer legislation on Second Order of Sexual

Harassment would improve bystanders' protection (social and legal protection) in order to intervene and to become upstanders [34].

One of the aims of the BraveNet model consists of its ability to be transferred to other spaces, meaning different territories and situations. From the legal side, the legal protection of those who suffer SOSH because of intervening is urgent and necessary. The pioneer SOSH legislation in one European Parliament contributes to such aim [34], inspiring other legislations in this sense.

The model we have developed is called Dialogic Scaling-Up Model for SOSH Prevention. The model and its implementation into a Platform we created is called BraveNet 0 violence: Upstander Social Network. The dialogic model shaped in its line is scientifically called "Contract on Dialogic Inclusion" [35], consisting in a process of several agents from the community, as stakeholders, researchers, social agents, involved to implement, in an entity, institution, and school, those successful practices that have proven to help in overcoming social and educational inequalities. In this contract, each sector has a role to be determined jointly agreed through dialogue that is set equal since that defines responsibilities and commitments with the shared goal of working towards social inclusion through the implementation of strategies for success.

Through the Dialogic Scaling-Up Model for SOSH Prevention, this model embodies a process that goes from Local to Regional and International levels, going from civil society to regional entities, to international movements, to policymakers and legislators. The cultural, social, and legal changes are needed to support the fight against domestic violence. This model will aim at (1) promoting social awareness on SOSH; (2) promoting social awareness on the need to socially and legally defend victims' supporters in order to eradicate DV; (3) building up the process through which we would raise awareness at local, regional, and EU level through the following pillars.

*3.6. The Dialogic Scaling-Up Model for SOSH Prevention Includes 4 Pillars*

1st pillar LOCAL level: it involves civil society, women entities, neighborhood associations . . . Professionals who assist women victims of DV, people who surround the victims (neighbors, relatives, people who may have some contact with them such as bakers, pharmacists . . . ), people who are around the upstanders. Scientific evidence will be transferred to them.

2nd pillar REGIONAL level: it involves the transferability of their commitment and demand to other platforms, such as NGOs, regional governments, entity federations, political parties, unions, national and European Ombudsman, schools, universities, companies . . .

3rd pillar EUROPEAN level with international support: European Women's Lobby, WAVE (Women Against Violence Europe), UN Women, EWLA (European Women Lawyers Association), ambassadors committed to the cause . . .

4th pillar POLICY WORKSHOPS: Drawing on the existing data on SOSH and the scientific evidence on how to prevent and eradicate this social problem, a set of evidence-based policy workshops will be organized to transfer this knowledge to policymakers and politicians.

Considering the above-mentioned pillars, these potential workshops will be organized taking these aspects into account: (a) Place; (b) Duration; (c) Audience: researchers on SOSH and DV, policymakers, stakeholders, MEP, end-users, members of civil society (women's associations, European Women's Lobby); (d) Methodology: Following EC's report on skills for evidence-informed policymaking [36] and the approach proposed by Marshall Ganz on Public Narratives [37], the policy workshops will be constructed on the framework of storytelling aimed at fostering policy change and sharing effective stories on the protection of protectors.

Thus, the BraveNet 0 violence: Upstander Social Network, will include (1) the "voices of citizenship": victims or their supporters will be able to present their own testimonies (like the #metoo hashtag which permitted thousands of survivors to break their silence),

but also to participate in the dialogue on the legislative demand, and how to gradually add implications, to achieve social impact. (2) The "voice of science" will be included (data about SOSH, scientific evidence on the impact of bystander interventions on reducing DV). (3) The Dialogic Scaling-up Model for SOSH Prevention.

## 4. Discussion

The situation of the pandemic and the lockdown caused by COVID-19 has also caused a health and social crisis that includes gender-based violence [38]. One of the problems that social entities have had to intervene in is a situation where many victims were confined with their aggressors [39]. Faced with this challenge, some entities have developed their own intervention program. In this article, we aim to know what type of entities were those that had intervened in situations of domestic violence during confinement, what type of interventions, if they had reprisals, and if they caused second-rate victims. Likewise, we wanted to see what impact the actions carried out had on the victims or potential victims, with the additional aim that these actions can be transferred to other contexts and thus protect the people who intervene. To do this, a questionnaire was prepared that seeks to measure the existence of second order of sexual harassment during confinement and the measures that help promote intervention and overcoming this potential harassment.

Tackling SOSH creates awareness about the protection not only of victims but also of those who dare to support them. Thus, more people would have access to achieve justice. In this sense, Dahanayake and colleagues [40] emphasize the discrimination in the very different contexts in which the second order of sexual harassment may occur, highlighting the importance of integrating justice and fairness standards when implementing programs of diversity management, arguing that social justice provides benefits for society. In this way, more people will be encouraged to intervene, both in case of witnessing a case of direct violence and SOSH violence. Actions in line with the bystander intervention have the objective of encouraging people to participate and to protect those who intervene.

Much progress has been made since the approval of the Title IX legislation in 1972 [41] which played a significant role in launching many complaints and campaigns against sexual violence in academia. Title IX legislation still serves for complaints about sexual violence within universities [42] and many countries have taken their example for their own legal advances regarding the protection of women's rights and the situations of violence they may suffer [43]. In the same sense, the pioneering approval of Second Order Violence [17,34] will open the way for other legislations to include protection for active bystanders, and above all, it will reduce the negative consequences of bystanders to intervene [44]. In fact, the purpose of the 2012 EU Directive on the rights of victims aims at making sure that any potential crime victim receives adequate information, support, and protection, and that they are able to participate in criminal proceedings and are subject to evaluation [45].

## 5. Conclusions

The current situation poses unprecedented social and individual challenges for which it is essential to offer solutions from science. Among several issues to be addressed, it is important to understand the mechanisms, spaces, or actions that help people in a situation of confinement to better cope with this situation and work from a plurality of professions to overcome the new challenges of COVID-19. People from different contexts are suffering from gender-based violence, sexist violence, which affects women and their children. A situation that has been aggravated by confinement, which for many women and minors has meant being 24 h with their abusers, without having other spaces or interactions. On the other hand, solidarity, the intervention of the community or of associations and entities, has become more necessary alternative than ever. Many people, entities, and associations have dedicated themselves to being the support that has been key for the survivors. Often this support involves attacks, criticism, and more violence. How to deal with or prevent these attacks has made a difference in the support and overcoming of this sexist violence.

That is why this study aims to deepen this impact so that these actions can reach more places around the world.

This article provides insights into knowing the reality of second order of sexual harassment during the confinement situation in the region of Catalonia, promoting more awareness about it, as well as actions that promote successful accompaniment for both the victims and their supporters [46]. The results showed that the member entities of the Unitary Platform intervened in cases of domestic violence during the confinement and did not suffer attacks for doing so in that period, although they did suffer them at another time [33]. It is clear that being part of the Platform, having support, and a strong network, encourages intervention. The same happened in other cases when grassroot movements have achieved social impact [47]. Therefore, as the support network is crucial, our contribution is made concrete in the BraveNet 0 violence: Upstander Social Network, which is a dialogical model, which aims to prevent SOSH, while pretending to be transferable to other contexts. Our analysis claims that a support network can have an impact in three terms:

(1)  Short-term: (a) To promote attitudes of support and an atmosphere of encouragement to all those who dare to support victims during COVID-19 and beyond. (b) To build awareness among people, neighbors, etc., by creating an emerging movement as a trigger for social change, while promoting the "social opportunity" needed a social claim to become a law. (c) To enable women to find security contexts standing with them, so they can break the silence, increasing their own and their children's security. (d) To improve scientific evidence on SOSH, available to citizens, politicians, and legislators, promoting social action and legal changes.

(2)  Medium-term: (a) To create a social context in favor of both victims and those who defend them. (b) To provide an institutional and community resource to have access to in order to find solutions to DV and to define common strategies to eradicate it. In other words, a place to go to find the pulse and create joint solutions for the DV problem for the victims' surroundings. (c) To increase the number of European citizens informed about this social claim and about the Dialogic Scaling-up Model for SOSH Prevention in order to be assumed in their contexts. (d) To impact on legislation mechanisms to include second order of sexual harassment across EU member states.

(3)  Long-term: (a) To legislate SOSH to eradicate DV deaths. (b) To transfer this impact to other areas where gender-based violence is present, such as schools, universities, companies, etc., enabling second order of sexual harassment to be legislated. (c) To contribute to SDG 5: Achieve gender equality and empower women and girls. (d) To foster a solidarity network at the European level against DV and SOSH to contribute to erasing any form of harassment and discrimination.

All in all, this research tackles the consideration of second order of sexual harassment as part of the legislation on gender-based violence recently approved by the Catalan Parliament. Victims need support in order to go forward and, in times of COVID-19 research, it is crucial in order to develop trustable information [48]. Any upstander can become a victim of SOSH because of giving their support. Therefore, it is crucial to contribute building awareness among all social actors, communities, and policymakers; creating that point of social awareness, which generates a social movement, creating the political opportunity; having legislators taking the issue of protecting not only direct victims but also victims of SOSH seriously. Thus, this research is contributing to inspiring other studies to be conducted across countries and cultures beyond the Catalonia region, to build a society standing with upstanders.

**Author Contributions:** Conceptualization, L.P. and P.M.; methodology, L.P.; software, M.S.; validation, A.V., M.S., and P.M.; formal analysis, L.P.; investigation, A.V.; resources, P.M.; data curation, A.V. and M.S.; writing—original draft preparation, A.V.; writing—review and editing, P.M.; visualization, M.S.; supervision, L.P. All authors have read and agreed to the published version of the manuscript.

**Funding:** This research received no external funding.

**Institutional Review Board Statement:** The study was conducted according to the guidelines of the Declaration of Helsinki of the World Medical Association, and approved by the Institutional Review Board (called Ethics Committee) of CREA (Center of Research on Excellence for All). Protocol code 20210228; date of approval 26 February 2021.

**Informed Consent Statement:** Informed consent was obtained from all the people involved in the study.

**Data Availability Statement:** The data presented in this article are available under request to the corresponding author. The data are not publicly available because of privacy issues and anonymity and of the participants.

**Conflicts of Interest:** The authors declare no conflict of interest.

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
