# Peer review of "BraveNet Upstander Social Network against Second Order of Sexual Harassment"

_sustainability, doi:10.3390/su13084135_

Round 1

Reviewer 1 Report

The authors do a great job at presenting the material and the design and layout is high. My main criticism is that the authors use acronyms quite frequently (e.g SOSH or GBV) and these could be introduced a bit clearer at the start.

The article needs to check its use of English and could benefit from having a good thorough grammar check. 

Section 1.1 tends to repeat itself throughout and I wonder if the points made could be restated or more examples given and possibly referenced? 

Section 2- I question the use of the work "materials" here. Could participants or profile of participants be used? Materials feels to objectify the individuals. 

Section 3.4 feels very skinny and I question if it could be expanded on somewhat.

3.5 Unitary Platform Membership- can this be further expanded on?  

lines 341-357: this paragraph could use some serious editing. It is not clear . and I would recommend some shorter sentences. 

Author Response

Response document:

REVIEWER 1

Comment 1:

The authors do a great job at presenting the material and the design and layout is high. My main criticism is that the authors use acronyms quite frequently (e.g SOSH or GBV) and these could be introduced a bit clearer at the start.

Response 1

Thank you for your comments Reviewer 1, and all the suggestions we have made. We reduced the use of acronyms throughout the paper and introduced SOSH and GBV at the beginning of the paper. Please see Pages 2, 3, and other occasions on which the terms appeared. At some point, we maintained the SOSH acronym as we consider it is important because it is known as such by the scientific community.

Comment 2:

The article needs to check its use of English and could benefit from having a good thorough grammar check. 

Response 2

Thank you for this suggestion. The article had a deep revision of English, including grammar check, throughout article. These changes can be observed in the main document in track changes.

Comment 3:

Section 1.1 tends to repeat itself throughout and I wonder if the points made could be restated or more examples given and possibly referenced? 

Response 3

To answer this comment we decided to put together section 1.1 and section 1.2 (see pages 2-3), as both provide information that can be related. We have removed some information and added some references in this new section 1.1.

Comment 4:

Section 2- I question the use of the work "materials" here. Could participants or profiles of participants be used? Materials feel to objectify the individuals. 

Response 4

Thank you for this. The word “materials” has been deleted in section 2. “Participants” is the word used for the first sub-section of this section 2 (see page 4).

Comment 5:

Section 3.4 feels very skinny and I question if it could be expanded on somewhat.

Response 5

We totally agree with your comment and 3.4. has been actually deleted as of a section, but unified with the previous one.

Comment 6:

3.5 Unitary Platform Membership- can this be further expanded on? 

Response 6

Section 3.5. has been expanded, see page 7.

Comment 7:

lines 341-357: this paragraph could use some serious editing. It is not clear. and I would recommend some shorter sentences. 

Response 7

Thank you for this. This entire paragraph has been reoriented and some changes have been shortened. See page 7.

Reviewer 2 Report

General conclusions: In general the issues of gender-based violence and the domestic violence still exist and are serious problems in the XXIst Century. The phenomenon of the violence is present in the European Union countries and have different aspects. In addition in some countries the politicians deny the need for the common anti-violence regulation (Istanbul Convention). For the mentioned reasons I think the article is interesting and worth to consider for the publication. Nevertheless I have some doubts about few aspects and opinion presented in the article.

  1. Introduction
  2. The Authors consider the gender based violence that affects the women but it must be remembered that the domestic violence also affect the men. In my opinion in the introduction part the Authors should stress that the domestic violence affect both men and women but according to the data more often it affects the women. For this reason the Authors decided to write about gender based violence on the women’s example. Without this notation the article is not the objective one, as it presents only the women as the victims.
  3. The article is about the specific situation of Covid-19 pandemic. The Authors write about those specific circumstances but in my opinion it would be useful to compare the pre-Covid-19 and the pandemic conditions of bystander interventions. For example I recommend to present a table in which the Authors compare characteristics of the conditions in the both situations.
  4. The Authors write about the bystanders intervention and the risk of reprisal against them. They examine it in the empirical part. Nevertheless they do not distinguish the formal and the informal bystanders that intervene in the situation of the domestic violence. The participants of the study are the official members of Unitary Platform against Gender Violence – so they are different kind of institutions. When they are asked about potential attacks for intervening, they respond negatively (lines 305-317). It could be the case that the institutions are prepared for this kind of situations but probably if the Authors examine  informal bystanders(friends, neighbors, family of the violence victims) the conclusion are totally different. So in my opinion there is a need to underline that the article focuses on the case of the official bystanders not the informal ones. Moreover it should be explained why only the formal entities are taking into consideration.
  5. Materials and Methods

The Authors conduct a questionnaire among the social entities from Catalonia Region. In my opinion it should be underlined that the article is a case study for this region and for particular entities.

  • Results
  1. The Authors present the entities’ scopes (lines 273-283) and then they present results of the study. I have a question if the Authors examined relations between the type of the entities and the type of the answers in the questionnaire? I can only find the simply presentation of the results without correlation analysis.
  2. Lines 380-396 (The Dialogic Scaling-up Model for SOSH) present so called 4 development steps of the model. In my opinion they are rather the pillars or the components of the model not the steps. The steps are the phases of the phenomenon and I do not agree that the first phase of the model could be the local one. Nobody can act if there are no social, political or legal support of the activities. So firstly we need the cultural, social, legal change that support the fight with the domestic violence.
  3. Conclusions

It must be stressed that the study was done only for the Catalonia region so there is a need to deepen the study and conduct it across the countries and cultures.

Author Response

Response document

REVIEWER 2

Comment 1:

General conclusions: In general the issues of gender-based violence and the domestic violence still exist and are serious problems in the XXIst Century. The phenomenon of the violence is present in the European Union countries and have different aspects. In addition in some countries the politicians deny the need for the common anti-violence regulation (Istanbul Convention). For the mentioned reasons I think the article is interesting and worth to consider for the publication. Nevertheless I have some doubts about few aspects and opinion presented in the article.

RESPONSE 1:

Thank you for your comment, we totally agree on the importance of addressing gender-based violence from the research perspective and the present paper contributes to this challenge, in line with the Istanbul Convention, by also introducing in the debate a new concept, the second order of sexual harassment.

Comment 2:

  1. Introduction
  • The Authors consider the gender based violence that affects the women but it must be remembered that the domestic violence also affect the men. In my opinion in the introduction part the Authors should stress that the domestic violence affect both men and women but according to the data more often it affects the women. For this reason the Authors decided to write about gender based violence on the women’s example. Without this notation the article is not the objective one, as it presents only the women as the victims.

RESPONSE 2:

Thank you for this. We did not specify to whom affects gender-based violence, but indeed, domestic violence may affect all genders, so we include this idea accordingly. (see page 2).

Comment 3:

  • The article is about the specific situation of Covid-19 pandemic. The Authors write about those specific circumstances but in my opinion it would be useful to compare the pre-Covid-19 and the pandemic conditions of bystander interventions. For example I recommend to present a table in which the Authors compare characteristics of the conditions in the both situations.

RESPONSE 3:

Thank you for this comment. Actually, this is a very interesting point. It was not the aim of this article to compare pre and during the covid pandemic, rather analyze which situations may limit bystanders to intervene. However, the paper studies the bystander intervention approach in general, as a phenomenon, which means, pre-Covid and then, throughout the survey, authors answered about the bystander intervention during the lockdown. To see the results of the survey’s question, see on page 6 (section 3.2. intervention in confinement). The conclusion states on the fact that people tend to intervene if they feel protected or supported by a network in the potential situation of being in pain.  

Comment 4:

  • The Authors write about the bystanders intervention and the risk of reprisal against them. They examine it in the empirical part. Nevertheless they do not distinguish the formal and the informal bystanders that intervene in the situation of the domestic violence. The participants of the study are the official members of Unitary Platform against Gender Violence – so they are different kind of institutions. When they are asked about potential attacks for intervening, they respond negatively (lines 305-317). It could be the case that the institutions are prepared for this kind of situations but probably if the Authors examine  informal bystanders(friends, neighbors, family of the violence victims) the conclusion are totally different. So in my opinion there is a need to underline that the article focuses on the case of the official bystanders not the informal ones. Moreover it should be explained why only the formal entities are taking into consideration.

RESPONSE 4:

Thank you for sharing this thought. Our objective when analyzing entities of the unitary platform against gender violence is to show the difficulty of intervening even from people who are part of an entity. An important element is that the representatives of the entities answered as individual persons, not as part of the entity. While it is true that there are different types of bystanders, and that the members of an entity may be more sensitive to the reality of sexual harassment, this article aims to provide a general idea of the intervention of the community, and especially of the support to those who support. Despite this, section 3.3. Potential attacks for intervening in supporting survivors (now they are other lines than those mentioned by the reviewer), have been modified accordingly. 

Comment 5:

  1. Materials and Methods

The Authors conduct a questionnaire among the social entities from Catalonia Region. In my opinion it should be underlined that the article is a case study for this region and for particular entities.

RESPONSE 5:

We already mentioned this point in subsection “participants” (page 4) part of the Methods section, starting on page 3.

Comment 6:

  1. Results
    • The Authors present the entities’ scopes (lines 273-283) and then they present results of the study. I have a question if the Authors examined relations between the type of the entities and the type of the answers in the questionnaire? I can only find the simply presentation of the results without correlation analysis.

RESPONSE 6:

Section 4.1. Entity scope (which is currently placed in different lines), has been placed on the “Results” part of this paper, as the description of each entity answering the questionnaire was made by the entities themselves, instead of by researchers describing the social entities. We did not correlate the results of the study with each entity type for several reasons: our aim was the overall intervention, the entities are actually not showing significant differences among each other, for anonymity reasons, and also because simply correlation is not so obvious to be established, considering than intervention policy depends on several factors rather than the type of the entity.

Comment 7:

  • Lines 380-396 (The Dialogic Scaling-up Model for SOSH) present so called 4 development steps of the model. In my opinion they are rather the pillars or the components of the model not the steps. The steps are the phases of the phenomenon and I do not agree that the first phase of the model could be the local one. Nobody can act if there are no social, political or legal support of the activities. So firstly we need the cultural, social, legal change that support the fight with the domestic violence.

RESPONSE 7:

Thank you for raising this point. We already introduced this consideration on page 8, specifying the pillars of the model in section: The Dialogic Scaling-up Model for SOSH.

Comment 8:

  1. Conclusions

It must be stressed that the study was done only for the Catalonia region so there is a need to deepen the study and conduct it across the countries and cultures.

RESPONSE 8:

We already introduced this point for the conclusion section (page 10). Thank you.
